# Modulating Heterologous Pathways and Optimizing Culture Conditions for Biosynthesis of *trans*-10, *cis*-12 Conjugated Linoleic Acid in *Yarrowia lipolytica*

**DOI:** 10.3390/molecules24091753

**Published:** 2019-05-06

**Authors:** Xun Wang, Qianjun Xia, Fei Wang, Yu Zhang, Xun Li

**Affiliations:** 1Jiangsu Provincial Key Lab for the Chemistry and Utilization of Agro-Forest Biomass, Nanjing Forestry University, Nanjing 210037, China; 18362929232@163.com (X.W.); jsxqj1991@163.com (Q.X.); hgwf@njfu.edu.cn (F.W.); yuzhang@njfu.edu.cn (Y.Z.); 2Jiangsu Key Laboratory of Biomass-Based Green Fuels and Chemicals, Nanjing Forestry University, Nanjing 210037, China; 3College of Chemical Engineering, Nanjing Forestry University, Nanjing 210037, China

**Keywords:** conjugated linoleic acid, metabolic engineering, *Yarrowia lipolytica*, *Propionibacterium acnes* isomerase, *Mortierella alpine* Δ^12^ desaturase

## Abstract

A novel recombinant strain has been constructed for converting glycerol into a specific conjugated linoleic acid isomer (*trans*-10, *cis*-12 CLA) using *Yarrowia lipolytica* as host. The lipid accumulation pathway was modified for increasing lipid content. Overexpression of the diacylglycerol transferase (DGA1) gene improved the intracellular lipid yield by approximately 45% as compared to the original strain. The corresponding intracellular lipid yield of recombinant strain WXYL037 reached 52.2% of the cell dry weight. In combination with integration of Δ12 desaturase from *Mortierella alpina* (MA12D) and DGA1, the linoleic acid (LA) production content reached 0.88 g/L, which was 2-fold that of the original strain. Furthermore, with overexpressed DGA1, MA12D and *Propionibacterium acnes* isomerase (PAI), the titer of *trans*-10, *cis*-12 CLA in WXYL037 reached 110.6 mg/L after 72 h of shake flask culture, representing a 201.8% improvement when compared with that attained in the WXYL030 strain, which manifested overexpressed PAI. With optimal medium, the maximum CLA content and lipid yield of *Y. lipolytica* Po1g were 132.6 mg/L and 2.58 g/L, respectively. This is the first report of the production of *trans*-10, *cis*-12 CLA by the oleaginous yeast *Y. lipolytica* using glycerol as the sole carbon source through expression of DGA1 combined with MA12D and PAI.

## 1. Introduction

Linoleic acid (LA, 18:2) and conjugated linoleic acid (CLA) have been attracting significant attention due to their potential health benefits, such as protection against carcinogenesis, reduction of atherosclerosis lesions, body fat modulation, anti hypercholesterolemic effects, and immunity enhancement with simultaneous attenuation of inflammation [1,2,3,4]. Compared to LA, CLA can lose electrons or hydrogen atoms to form free radicals more rapidly, which makes it more susceptible to oxidation reactions and more biologically active [5]. CLA is a family member of a mixture of positional and geometric isomers of LA in which the two double bonds are conjugated. In nature, CLA is generally found in meat and dairy products [6]; however, the concentration in foodstuffs is insufficient for any therapeutic application or nutritional requirements.

A sustainable alternative is to produce CLA directly from abundant and renewable resources through metabolic engineering and synthetic biology approaches, which presents numerous advantages such as a short preparation cycle, less labor requirement, reduced impacts of place, season and climate compared to others methods [7]; however, one of the challenges is developing a microbial catalyst for high product yields from inexpensive raw materials. A number of microorganisms have been identified as potential CLA producers. However, the levels of CLA production are relatively low [8,9]. It is well known that CLA is generally produced from LA through LA isomerase. The cytosolic LA isomerase derived from *Propionibacterium acnes* (PAI) which can produce *t*10, *c*12-CLA is the only isomerase that has been successfully expressed in multiple plants and microorganisms [10,11]. However, the high price of LA limits its use as a raw material for CLA preparation. In the previous study, vegetable oils, which are rich in oleic acid and LA, have been proven useful as feedstocks for CLA production. Synergistic catalytic systems based on immobilized PAI and *Rhizopus oryzae* lipase (ROL) were developed for converting plant oil into CLA, and the corresponding conversion ratio of LA to CLA was 90.5% [11,12]. Nevertheless, the high cost of enzyme preparation limits the utility of this process. 

Meanwhile, as an oleaginous yeast, *Yarrowia lipolytica* has been considered as a potential candidate as a platform organism for superior CLA production, owing to its high lipid production rate (approaching 90% of cell dry weight) with a high proportion of oleic acid (OA, 18:1) and LA in the fatty acid composition and clear genetic background [13,14,15]. The intracellular lipids of *Y. lipolytica* can be used as raw materials for biodiesel production and can also be used for edible oil production as common vegetable oil because of its food safety. The intracellular lipids of this yeast are primarily OA and LA, of which OA accounts for approximately 50% of the total fatty acid content and LA accounts for nearly 15–20% of the total fatty acid content [14,16]. CLA production can be greatly increased by converting OA into LA. LA can be synthesized from OA through Δ^12^ desaturase from *Mortierella alpina* (MA12D).

Due to its well-studied lipid metabolism, the significant array of genetic tools and a fully sequenced genome, *Y. lipolytica* has become a model oleaginous yeast for the production of lipids and lipid-derived biofuels. The use of the oleaginous yeast *Y. lipolytica* benefits from its well-developed tools for engineering the lipid metabolic pathway, and this species can transform inexpensive raw glycerol derived from biodiesel production into high-value products such as CLA. In addition to glucose or fats, raw glycerol can serve as the sole carbon for *Y. lipolytica* generation of a large amount of high value-added chemicals, such as gamma-LA, ricinoleic acid and omega-3-eicosapentaenoic acid (EPA) [17,18,19]. Meanwhile, global biodiesel production in 2014 was approximately 29.7 billion liters, and the primary byproduct glycerol, which accounts for nearly 10% of biodiesel production, is as affordable as $0.24 per kilogram [20]. Synthesis from glycerol to triacylglyceride (TAG) by *Y. lipolytica* follows the Kennedy pathway [21]. First, the glycerol-3-phosphate (G3P) molecule is either synthesized from dihydroxyacetone phosphate (DHAP), catalyzed by NAD^+^-dependent glycerol-3-phosphate dehydrogenase (GPD1) [22], or derived from glycerol via glycerol kinase (GUT1). Then, G3P is acylated by G3P acyltransferase (SCT1) and catalyzed by LPA acyltransferase (SLC1) to yield phosphatidic acid (PA). After that, diacylglycerol (DAG) is generated through dephosphorylation of PA by phosphatidic acid phosphohydrolase (PAP). In the final step, TAG is produced either by diacylglycerol transferase (DGA1), with acyl-CoA as an acyl donor, or by phospholipid DAG acyltransferase (LRO1), with phospholipid (PL) as an acyl donor (Figure 1). In addition, the TAG accumulated in cells can be further hydrolyzed into glycerol and free fatty acids (FFA) by endogenous lipases to provide raw materials for the synthesis of additional products while also providing energy for cell growth [23]. PAI catalyzes the isomerization of LA to *t*10, *c*12-CLA [24]. Zhang et al. reported that the highest titer of *t*10, *c*12-CLA, 45 mg/L, was obtained in genetically modified *Y. lipolytica* by introducing PAI and MA12D using YPD medium in shaking flask cultures after 72 h cultivation [25]. During this study, in order to further improve the contents of lipid and CLA in *Y. lipolytica* from non-oil substrates, the copy numbers of pivotal genes, GPD1 or DGA1, were increased, this provides an enhanced driving force towards production of lipid. Subsequently, raw glycerol was utilized as carbon source to overproduce CLA in *Y. lipolytica* by modifying the lipid accumulation pathway and heterologously expressing MA12D and PAI.

## 2. Results and Discussion

### 2.1. Effects of DGA1 and GPD1 on Lipid Production in Y. lipolytica

Biosynthesis of lipids and their derivatives is a complex and tightly regulated metabolic undertaking. More and more literature has reported that the *DGA1* and *GPD1* genes serve important roles in lipid accumulation. Recently, Tai et al. have shown that overexpression of DGA1 yielded a 4-fold increase in lipid production over control [16]. In this study, to implement the identified target, the copy numbers of *DGA1* and *GPD1* were increased in the *Y. lipolytica* genome, respectively, using a constitutive hp4d promoter which had been widely used in recent years [26,27,28,29]. For the rapid screening of high-yield oleaginous yeast, WXYL010, WXYL015 and Po1h were stained with Nile Red and their fluorescence intensity at 520 nm compared (Appendix A). In a lipid-rich environment, Nile Red can emit intense fluorescence with varying colors from deep red (for polar membrane lipids) to strong yellow-gold emission (for neutral lipids within intracellular storages). The results indicated that enhancing some key genes which link to the accumulation of lipids has effectively improved the lipid content in *Y. lipolytica*. Next, the effects of overexpression of DGA1 and GPD1 on lipid production in WXYL010 and WXYL015 strains were assessed (Figure 2). With glycerol as the sole carbon substrate, the wild strain Po1h produced 1.86 g/L lipid. It was found that the lipid titer of recombinant strains WXYL010 and WXYL015, which harbor DGA1 or GPD1 genes, both exceeded the wild strain Po1h, accumulating 2.60 g/L and 1.95 g/L lipid, respectively. Moreover, compared to the WXYL015 strain, the WXYL010 strain achieved higher titer values demonstrating up to 40% increases in production over the wild strain. It is probable that glycerol, as the sole carbon source, can be fully utilized by the glycerol kinase (Gut1) instead of GPD1, which can only convert DHAP to G3P. It is worth noting that significant increases of lipid content have been achieved when overexpressing DGA1, which exerts a direct catalytic effect on DAG to form TAG. These results demonstrated that increased *DGA1* copy number can enhance metabolite levels and subsequently improve the efficiency of lipid production.

### 2.2. Establishment of a LA Biosynthesis Pathway in Y. lipolytica

The free fatty acids in *Y. lipolytica* are primarily oleic acid and linoleic acid. Sun et al. reported a recombinant strain which produced γ-LA at titers of 71.6 mg/L through heterologous expression of a heterologous desaturase gene [30]. In this study, to increase the proportion of unsaturated LA in lipids, therefore the *MA12D* gene was introduced into *Y. lipolytica* Po1g, resulting in the WXYL020 strain. It was found that the transformant indeed showed a significant increase in LA content. The content of LA was increased from 26% to 34%, eventually reaching 0.58 g/L, and it was interesting to note that the content of methyl palmitoleate (C16:1) was also slightly enhanced; however, the methyl stearate (C18:0) content was decreased (Figure 3A). This could be caused by various side reactions of Δ12 desaturase with stearic acid and palmitic acid and the dehydrogenation of oleic acid. It is speculated that this enzyme may catalyze the dehydrogenation of stearic acid and palmitic acid to oleic acid and palmitoleic acid, respectively. The increased copy number of the *DGA1* gene in the genome of *Y. lipolytica* provides an excellent platform for high yields of lipid production. Therefore, *DGA1* and *MA12D* were expressed in tandem within the WXYL025 strain: the content of LA was maintained at 34%, and the lipid accumulation reached up to 50% of its dry cell weight (DCW), at which time the LA yield reached 0.88 g/L (Figure 3B). The result showed that co-expression of two genes, *DGA1* and *MA12D*, further improved the content of LA on the foundation of high lipid titers.

### 2.3. Establishment of a CLA Biosynthesis Pathway in Y. lipolytica

As mentioned in the introduction, CLA has gained widespread attention. Recently, Zhang et al. have proven that immobilized PAI, which is the only isomerase successfully expressed in multiple plants and microorganisms, can efficiently convert LA into a specific CLA (t10, c12-CLA) [11,12]. In this study, to achieve higher added-value products, the *PAI* gene was expressed in *Y. lipolytica* Po1h, resulting in the WXYL030 strain.

As a result, the CLA has emerged and accounts for 3.3% of intracellular lipids, with a yield of 54.8 mg/L (Figure 4). Comparing *Y. lipolytica* Po1h strain with WXYL030, there is no significant change in the relative content of other lipid components. Meanwhile, hp4d-*PAI* (pWX030) was used for the tandem gene construction of *PAI+MA12D* (pWX034) and *PAI*+*MA12D*+*DGA1* (pWX037), respectively. The recombinant strains WXYL034 and WXYL037 both exceeded WXYL030, accumulating 68.2 mg/L and 110.6 mg/L of CLA, respectively. The CLA yield of the WXYL034 strain is approximately one-quarter more than that of WXYL030; at the same time, the content of methyl palmitoleate (C16:1) was also slightly enhanced, and methyl stearate (C18:0) was decreased compared to wild-type strains. This phenomenon further proves the hypothesis that oleic acid dehydrogenase exerts certain side effects on stearic acid and palmitic acid. The WXYL037 strain coexpression of *PAI*, *MA12D* and *DGA1* genes ultimately obtained the maximum CLA yield of 110.6 mg/L, which is the highest output of CLA by *Y. lipolytica* yet reported using glycerol as the sole carbon substrate.

### 2.4. Optimization of Flask Culture Conditions

#### 2.4.1. Effect of Carbon and Nitrogen Sources on CLA Production

To investigate the effect of the carbon source on lipid production, glucose, different grades of glycerol and waste oil were assessed. Altogether, the results indicate that WXYL037 using waste oils as the carbon source exhibit higher lipid formation/DCW (accumulated 0.56 g lipid/g DCW). Considering the instability and complex components of waste oil sources, waste oils were not chosen as a suitable carbon source of WXYL037. Meanwhile, lipid formation/DCW of WXYL037 on glycerol or crude glycerol, which was 0.49 g lipid/g DCW, were similar to those obtained on glucose (data not shown). Furthermore, the industrial production of glucose mainly depends on the hydrolysis of starch, which is not enough to meet the human needs of food production. In comparison, the production of raw glycerol has increased significantly due to its status as the principle side-product of biodiesel. The uniformity of the production process of biodiesel results in nearly the same composition of crude glycerol [31]. The development of processes for converting crude glycerol into higher value products is expected to make biodiesel production more economical [32]. Therefore, raw glycerol was chosen as the sole carbon source in this work. 

The growth of oleaginous yeast usually passes through the cell proliferation phase, the oil accumulation period, and the citric acid accumulation period. For *Y. lipolytica*, exogenous genes are generally expressed towards the end of the logarithmic growth phase of cell growth, in which case a large amount of nitrogen source is required. Similarly, the nitrogen source is also one of the essential components for microbial growth. For obtaining the highest CLA contents, ammonium chloride, ammonium sulfate, ammonium nitrate, tryptone, bacterial peptone, and soy peptone were selected as nitrogen sources (Figure 5A). It is obvious that organic nitrogen sources led to greater amounts of obtained biomass and lipid contents than inorganic nitrogen sources in *Y. lipolytica*. Simultaneously, the biomass and CLA contents obtained by the use of three kinds of organic nitrogen sources are similar, the CLA yield were about 109.8 mg/L, and the most inexpensive organic nitrogen source soy peptone was ultimately selected. 

#### 2.4.2. Effect of C/N Ratio on CLA Production

During CLA accumulation, the carbon to nitrogen mass ratio (C/N) in the medium represents an important factor affecting the CLA yield [13,33]. The C/N ratio has already been investigated as a key factor for CLA accumulation, and it is worth noting that the cells are primarily based on their own proliferation when the nitrogen supply in the medium is rich. Instead, the cells may enter the apoptotic phase in advance when the nitrogen composition is insufficient. As shown in (Figure 5B), the highest CLA production yield was achieved at the C/N ratio of 50:1, reaching 119.1 mg/L, which was improved by approximately 54% as opposed to that at the C/N ratio of 25:1. 

#### 2.4.3. Effect of CaCl_2_ Concentration on CLA Production 

Inorganic salts are required during microbial growth to promote the activity of enzymes and maintain the normal activities of the microorganisms for obtaining a large accumulation of lipids and CLA [34]. Therefore, the effects of varying concentrations of CaCl_2_ (0~0.5 g/L) were tested (Figure 5C). The total amount of CLA reached a maximum (128.3 mg/L) at concentrations of 0.2 g/L CaCl_2_, while lower or higher concentrations caused a decrease of yield.

### 2.5. CLA Production by Recombinant Strain 

To further determine the ability of the engineered strain to produce CLA in high yield, shake-flask culture was carried out using the engineered *Y. lipolytica* Po1g strain simultaneously integrated with *DGA1*, *MA12D* and *PAI* genes. Based on the above data, the most suitable culture conditions for CLA production using the WXYL037 engineered strain were 0.2 g/L CaCl_2_, glycerol and soy peptone as the carbon and organic nitrogen sources, and the C/N ratio of 50:1. As shown in (Figure 5D), under optimal conditions, the cell concentration increased rapidly within 0~30 h, which represented the logarithmic growth phase of yeast. After 30 h, yeast cells entered a stable growth phase. CLA is generally produced at the end of logarithmic growth, and the titer of CLA reached 132.6 mg/L and 5.2% of total fatty acid yield in transformed *Y. lipolytica* at 84 h and the conversion efficiency of glycerol to CLA (gram to gram) reached 0.66%. In view of these observations, the research provides an excellent strategy for the commercialization of inexpensive glycerol into higher value products by engineering the oleaginous yeast *Y. lipolytica*.

Despite the extensive progress made regarding CLA production in *Y. lipolytica*, many possible improvements can be achieved in enhancing CLA production. One approach is to adjust the metabolic pathway, such as through the deletion of glycerol-3-phosphate dehydrogenase gene (*GUT2*) or integration of additional key lipid synthesis genes into the *Y. lipolytica* genome by clustered regularly interspaced short palindromic repeats (CRISPR)/Cas9 technology to elevate the yield of lipids or free fatty acids [17,35].

Another possibility is further employing a chromosome integration technique to strengthen the copy number and transcription intensity of recombinant genes [25]. The used plasmid pINA1312 is a single-copy integrated plasmid, while the integrative *Y. lipolytica* vectors offer the possibility of multiple integrations and of a correlated increase in gene expression. Furthermore, the expression of the heterologous gene were driven by the constitutive hp4d promoter, which consists of regulatory element 4 *UAS1B and core promoter LEU2. There is a linear relationship between the number of UAS1B regulatory elements and the regulation intensity of promoter. The more UAS1B inserted upstream of LEU2, the higher the expression of proteins regulated by promoter [36]. Based on the above reasons, the subsequent work would focus on strengthening the copy number and transcription intensity of key recombinant genes (*PAI* and *MA12D*).

Finally, the concepts of optimization of high-density fermentation towards the final product are prominent strategies for the generation of desirable products in industrial applications. Zhang et al. reported that total CLA production in the fermenter using YNBD-SO medium (2% dextrose and 2% soybean oil as carbon source) was almost 4 g/L, about 90 times higher compared to that using YPD medium in shaking flask cultures (45 mg/L). The primary reason for the high CLA production of the *Y. lipolytica* strain may lie in the YNBD-SO medium resulting from the converting of cosubstrate soybean oil into CLA by PAI and lipase [12]. Besides, optimization of high-density fermentation using medium with proper oil supplement towards CLA are prominent strategies for the higher production of CLA. 

## 3. Conclusions

CLA has attracted significant attention due to anticarcinogenic and lipid/energy metabolism-modulatory effects; however, the concentration in foodstuffs is insufficient for any therapeutic application to be implemented. In this study, through refining lipid accumulation, integrating overexpression of diacylglycerol transferase (DGA1), Δ12 desaturase (MA12D) and *Propionibacterium acnes* isomerase (PAI), and optimizing culture conditions (carbon and nitrogen source, carbon to nitrogen mass ratio, CaCl_2_ content), the maximum achieved CLA content and lipid yield of *Y. lipolytica* Po1g were 132.6 mg/L and 2.58 g/L, respectively. This is the first report of the production of *trans*-10, *cis*-12 CLA by the oleaginous yeast *Y. lipolytica* using glycerol as the sole carbon source through expression of DGA1 combined with MA12D and PAI. This is the first report on CLA synthesis and the highest obtained CLA titer using glycerol as the sole carbon source in *Y. lipolytica* in shake-flask cultures. Therefore, an alternative production system for CLA from inexpensive and renewable sources of glycerol through metabolic engineering methods in *Y. lipolytica* has been demonstrated. The novel CLA synthesis pathway had two distinct characteristics: a higher CLA titer and utilization of inexpensive raw glycerol. It was difficult to obtain CLA because of its low concentration and complex composition in foodstuffs. However, crude glycerol was inexpensively and easily obtained from biodiesel production. Thus, it is meaningful to reconstruct a recombinant strain for CLA production from crude glycerol. In this study, a recombinant strain was constructed for enhanced lipid accumulation and efficient CLA production from crude glycerol. Simultaneously, a complete CLA synthesis pathway was developed, which could be widely used for the synthesis of additional value-added fatty acid derivatives. In conclusion, a high value-added compound CLA that can be produced from glycerol by *Y. lipolytica* is shown to represent a valuable contribution to the development of a cost-effective fermentation method based on renewable resources.

## 4. **Materials and Methods**

### 4.1. Plasmids and Strains

All strains and plasmids constructed in this work are listed in Table 1. The primers used for genetic modifications are listed in Appendix A. Standard protocols were followed for DNA manipulation, and all constructed plasmids were verified by sequencing [37]. The *Y. lipolytica* strains used in this study were derived from the wild-type strain *Y. lipolytica* W29 (ATCC 20460). The auxotrophic Po1g (Leu^−^) and Po1h (Ura^−^) used in all transformations were kindly provided by Prof. Catherine Madzak (Institute National de la Recherche Agronomique/AgroParisTech, Paris, France). All cloning procedures were carried out in *E. coli* strain Top10 (Thermo Fisher Scientific, Waltham, MA, USA). The genes *DGA1* (GenBank No. YALI0E32769g) and *GPD1* (GenBank No. YALI0B02948p, Appendix A) were amplified from *Y. lipolytica* Po1h’s genomic DNA by the Rapid Yeast Genomic DNA Isolation Kit (Sangon Biotech., Shanghai, China). The *DGA1* was flanked by *Pml* I and *Kpn* I restriction sites at the 5′ and 3′ ends, respectively and was amplified by PCR using PrimerSTAR MAX DNA polymerase (Takara Biotech, Co., Ltd., Dalian, China). After digestion, the fragments were ligated at the *Pml* I/*Kpn* I sites of the pINA1312 expression vectors to yield the pINA1312-DGA1 and pINA1312-GPD constructs, respectively.

The codon-optimized *MA12D* with the sequence GCCACA added before the start codon to create a Kozak sequence [38] was chemically synthesized and cloned into pINA1267, yielding the pWX020 plasmid. The *DGA1* gene and a modified hp4d promoter [26] were inserted downstream of *MA12D* in pWX020, yielding the pWX025 plasmid. The gene encoding PAI was cloned from the pET20b-PAI plasmid, which was preserved in laboratory, and cloned into pINA1312 [12], yielding the pWX030 plasmid. The DNA cassette containing the hp4d promoter and the *MA12D* gene were inserted downstream of the *PAI* gene in pWX030, yielding the pWX034 plasmid. In the same way, the DNA cassettes containing *MA12D* and *DGA1* genes, with each possessing the hp4d promoter were inserted downstream of the *PAI* gene in pWX030, yield the pWX037 plasmid.

As described previously, the engineered yeast strains were constructed via transformation of the corresponding plasmid, which was linearized with *Not*I and chromosomally integrated into Po1h or Polg strains, respectively, according to the lithium acetate method [39]. Transformants were plated on selective media and verified by PCR of prepared genomic DNA. 

### 4.2. Growth and Culture Conditions

*E. coli* strains were grown at 37 °C in low Luria-Bertani (LLB) medium (10 g/L tryptone, 5 g/L yeast extract, 5 g/L NaCl and pH 7.0) containing ampicillin (100 µg/mL). Media and growth conditions for *Y. lipolytica* have been described by Barth and Gaillardin [42]. Rich medium (YPG) was prepared with 20 g/L peptone, 10 g/L yeast extract, and 20 g/L raw glycerol. Fermentation medium, containing 50 g/L glycerol, 1.0 g/L peptone, 3.0 g/L MgSO_4_·7H_2_O, 0.2 g/L CaCl_2_ and 4.0 g/L KH_2_PO_4_, is used for lipid and CLA production. Cultivation was typically performed as follows: from the YPG plate, single colonies were inoculated into 5 mL YPG medium at 180 rpm and 28 °C for 48 h. A 0.5 mL seed liquid was inoculated into 50 mL YPG medium at 180 rpm and 28 °C for 48 h. For lipid production, 10% (*v*/*v*) seeds were inoculated into the fermentation medium. The cells were harvested for dry cell mass and lipid measurement. The growth state of *Y. lipolytica* was analyzed by measuring the optical density at a wavelength of 600 nm (OD_600_) using a Bio photometer Plus apparatus (Eppendorf, Hamburg, Germany). 

### 4.3. Lipid Extraction and Quantification

The *Y. lipolytica* cells were washed twice with distilled water and harvested by centrifugation at 10,000× *g* for 1 min. Then, 5 mL of 4 mol/L hydrochloric acid solution was added into the precipitation, and the mixture was retained at room temperature for 20 min, boiled for 5 min and rapidly cooled to −20 °C. Total lipids were extracted twice with 2:1 *v/v* chloroform-methanol (10 mL). The upper layer was collected, and the solvent was evaporated using a nitrogen streaming instrument (YueXu Technology, Nanjing, China). The sample was converted into fatty acid methyl esters (FAMEs) in 10% (*v*/*v*) hydrochloric acid-methanol solution at 65 °C for 3 h. Subsequently, the methanol was removed by rotary evaporation under vacuum. FAMEs were extracted by hexane and saturated NaCl solution. The organic residue was collected and stored at −20 °C until use. The residual water was removed using anhydrous Na_2_SO_4_.

FAMEs analysis was performed with a GC-7890A (Agilent, Santa Clara, CA, USA) equipped with a PEG-20M capillary column (0.32 mm × 30 m × 0.25 μm) and a flame-ionization detector (FID). Nitrogen was used as the carrier gas, and the total gas flow rate was 2 mL/min. The samples were measured with a split ratio of 15:1 with the injector and detector temperatures set to 205 °C and 230 °C, respectively. The column temperature was maintained at 120 °C for 1 min, then increased to 190 °C at a rate of 10 °C/min, further raised to 205 °C at a rate of 1 °C/min, and then maintained at 205 °C for 3 min. Methyl esters of LA and *t*10, *c*12-CLA were identified by comparisons with authentic FAMEs standards (CLA methyl ester, Sigma, St. Louis, MO, USA). Heptadecanoic acid methyl ester (Sigma-Aldrich, St. Louis, MO, USA) was used as an internal standard. Total lipid content was calculated as the sum of total fatty acid contents for five FAMEs: methyl palmitate (C16:0), methyl palmitoleate (C16:1), methyl stearate (C18:0), methyl oleate (C18:1), and methyl linoleate (C18:2). The LA concentration in plant oils is defined as the percentage of LA methyl esters versus the total amount of FAMEs in the plant oil. CLA yields were calculated using the follow equation: CLA conversion ratio (%) = (content of CLA/content of lipid) × 100%. CLA yield (mg/L) = (content of lipid × CLA conversion ratio)/volume of culture broth. 

### 4.4. Fluorescence and Electron Microscopy 

To screen high yield oil producing strains, WXYL010 and WXYL015 cells were diluted to OD_600_ = 1.0 by 8% (*v*/*v*) DMSO aqueous solution and stained with Nile Red dye (1.2 μg/mL) for 10 min in darkness. Samples were activated under 488 nm light, and fluorescence emission spectra of samples were analyzed under 520 nm irradiation by a Cytation3 cell imaging microplate detector (BioTek, Winooski, VT, USA). An Olympus IX71 fluorescence microscope (Olympus Optical Industrial, Tokyo, Japan) was used to capture fluorescence images with a 100 ~ 1000 × oil-immersion objective.

### 4.5. Statistical Analysis

Each joint biotransformation reaction was carried out in triplicate. Data were expressed as the mean ± SD. The data were subjected to one-way ANOVA or *t*-tests using SPSS 10.0 statistical software (SPSS, Chicago, IL, USA). *p* Values of <0.05 were considered statistically significant.

## Figures and Tables

**Figure 1 molecules-24-01753-f001:**
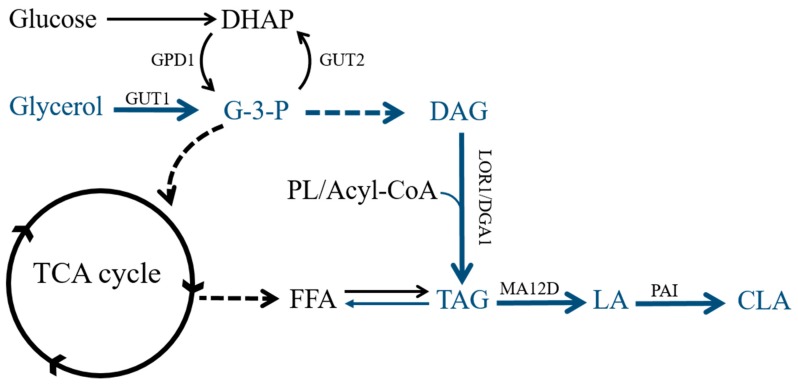
Overview of the lipid metabolism pathway for *Yarrowia lipolytica*. DHAP: dihydroxyacetone phosphate; G3P: glyceraldehyde 3-phosphate; DAG: diacylglycerol; PL: phospholipid; TAG: triacylglycerol; FFA: free fatty acids; LA: Linoleic acid; CLA: conjugated linoleic acid; GPD1: NAD^+^-dependent glycerol-3-phosphate dehydrogenase; GUT1: glycerol kinase; GUT2: glycerol-3-phosphate dehydrogenase; DGA1: diacylglycerol transferase; MA12D: *Mortierella alpine* Δ12 desaturase; PAI: *Propionibacterium acnes* isomerase.

**Figure 2 molecules-24-01753-f002:**
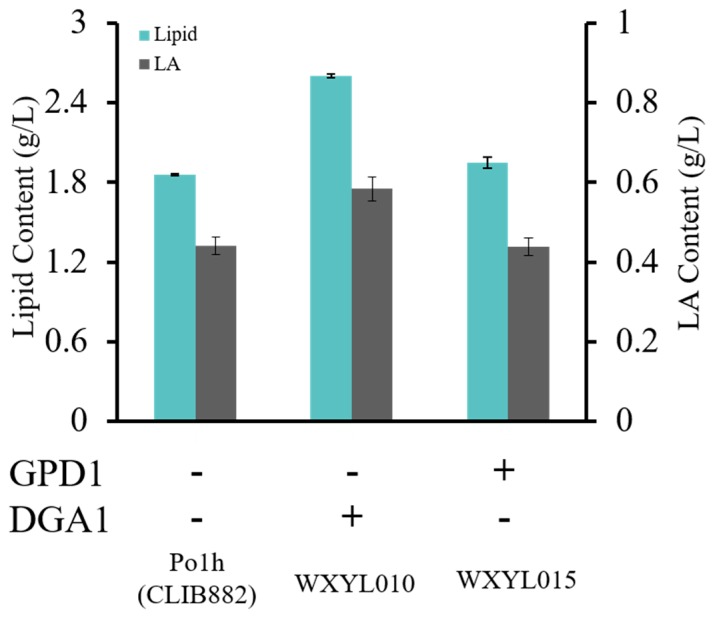
Analysis of strains overexpressing *GPD1* or *DGA1*. Lipid and LA contents of strains were analyzed after 72 h of culture (C/N mass ratio of 15). Lipid samples were performed in triplicate.

**Figure 3 molecules-24-01753-f003:**
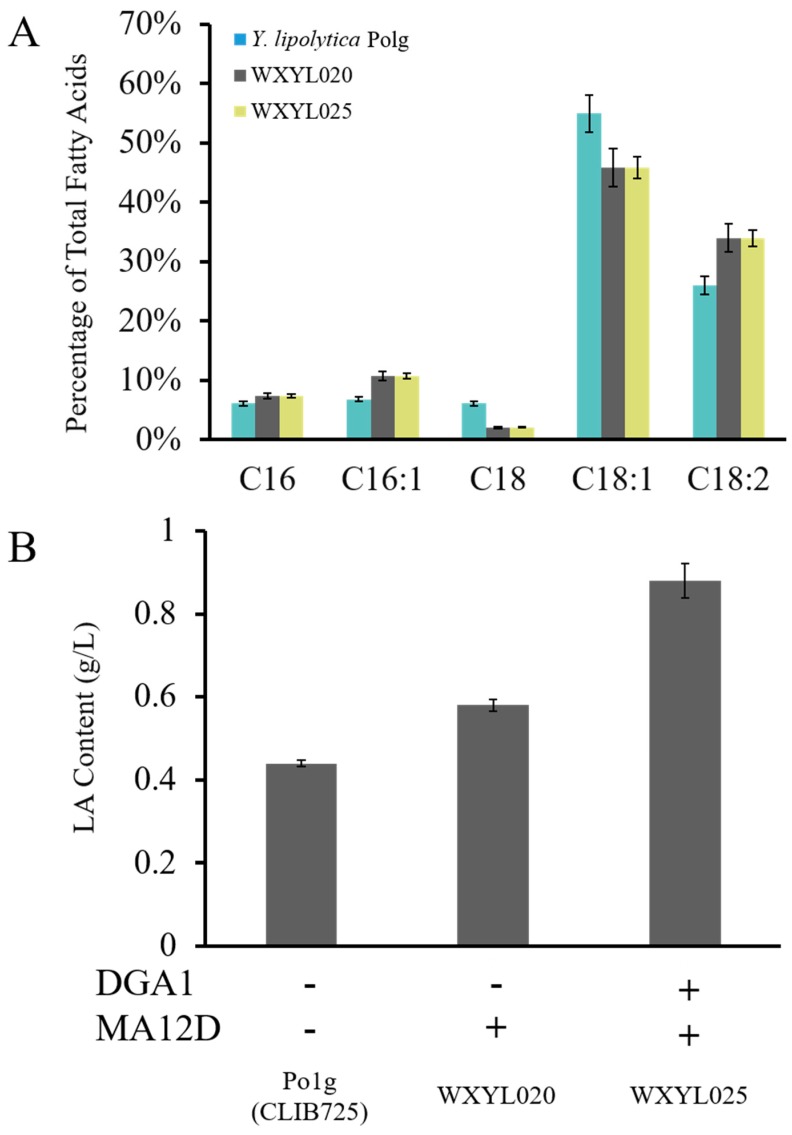
(**A**) Fatty acid distribution comparisons for WXYL020 and control strain wild *Y. lipolytica* Po1g grown in shake flasks. C16 methyl palmitate; C16:1 methyl palmitoleate; C18 methyl stearate; C18:1 methyl oleate; C18:2 methyl linoleate. (**B**) Analysis of strains overexpressing DGA1 or/and MA12D. LA contents of strains were analyzed after 72 h of culture. Lipid samples were performed in triplicate.

**Figure 4 molecules-24-01753-f004:**
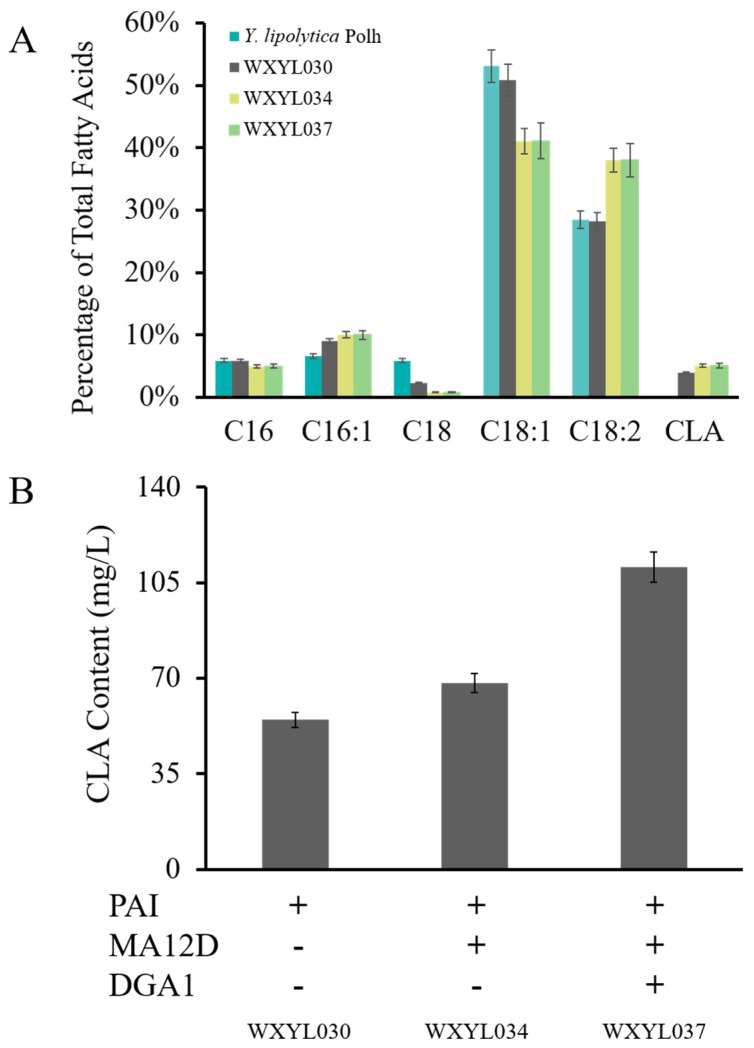
(**A**) Analysis of strains overexpressing PAI or/and MA12D, DGA1. Fatty acid distribution comparisons for WXYL030, WXYL034 and WXYL037 strain grown in shake flasks. (**B**) Conjugated CLA contents of strains were analyzed after 72 h of culture. Lipid samples were performed in triplicate.

**Figure 5 molecules-24-01753-f005:**
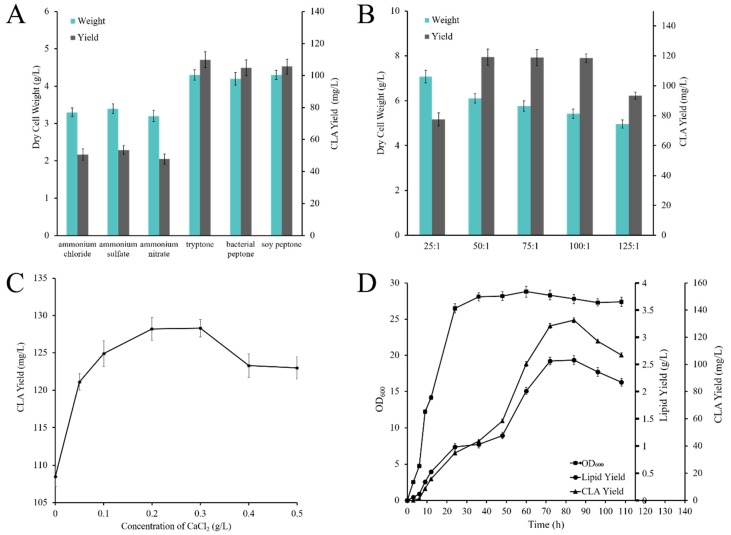
Effects of different culture conditions on *Y. lipolytica*. (**A**,**B**) Effects of different nitrogen sources and C/N ratios on CLA production. (**C**) The *Y. lipolytica* was cultured at different inorganic salt concentrations: Ca^2+^. (**D**) Under optimal conditions, the lipid and CLA production by recombinant strain WXYL037.

**Table 1 molecules-24-01753-t001:** Strains and plasmids used in this study.

Strains (Host Strain)	Genotype or Plasmid	Source or Reference
*E. coli*		
Top10	F^-^*mcr*A Δ(*mrr*-*hsd*RMS-*mcr*BC) φ80 *lac*Z Δ*M15* Δ*lac*X74 *rec*A1 *ara*Δ139 Δ(*ara-leu*)7697 *gal*U *gal*K *rps*L (Str^R^) *end*A1 *nup*G	Invitrogen
Plasmid		
pINA1312	hp4d, *XPR2*t, *ura3d4,* Kan^R^	[40]
pINA1267	hp4d, *XPR2* prepro, *XPR2*t, *LEU2*, Amp^R^	[41]
pWX010	pINA1312 hp4d-*DGA1*	This work
pWX015	pINA1312 hp4d-*GPD1*	This work
pWX020	pINA1267 hp4d-*MA12D*	This work
pWX025	pWX020 hp4d-*DGA1*	This work
pWX030	pINA1312 hp4d-*PAI*	This work
pWX034	pWX030 hp4d-*MA12D*	This work
pWX037	pWX034 hp4d-*DGA1*	This work
*Y. lipolytica*		
Po1g (CLIB725)	MatA, leu2-270, ura3-302::URA3, xpr2-332, axp1-2, Leu^−^, ΔAEP, ΔAXP, Suc^+^, pBR322	[26,41]
Po1h (CLIB882)	MatA, ura3-302, xpr2-232, axp1-2, Ura^−^, ΔAEP, ΔAXP, Suc^+^	[36,41]
WXYL010	MatA, ura3-302, xpr2-232, axp1-2 hp4d-*DGA1*-URA3	This work
WXYL015	MatA, ura3-302, xpr2-232, axp1-2 hp4d-*GPD1*-URA3	This work
WXYL020	MatA, leu2-270, ura3-302::URA3, xpr2-332, axp1-2 hp4d-*MA12D*-LEU2	This work
WXYL025	MatA, leu2-270, ura3-302::URA3, xpr2-332, axp1-2 hp4d-*MA12D*+hp4d-*DGA1*-LEU2	This work
WXYL030	MatA, ura3-302, xpr2-232, axp1-2 hp4d-*PAI*-URA3	This work
WXYL034	MatA, ura3-302, xpr2-232, axp1-2 hp4d-*PAI*+hp4d-*MA12D*-URA3	This work
WXYL037	MatA, ura3-302, xpr2-232, axp1-2 hp4d-*PAI*+hp4d-*MA12D*+hp4d-*DGA1*-URA3	This work

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
