# Peer review of "Modulating Heterologous Pathways and Optimizing Culture Conditions for Biosynthesis of trans-10, cis-12 Conjugated Linoleic Acid in Yarrowia lipolytica"

_molecules, 2019, doi:10.3390/molecules24091753_

Round 1
Reviewer 1 Report
The manuscript by Want et al. presents the heterologous expression of trans-10, cis-12 conjugated linoleic acid (CLA) in Yarrowia lipolytica using glycerol as the sole carbon source. They firstly overexpressed diacylglycerol transferase (DGA1) and Δ12 desaturase (MA12D) in the original strain, which improved 2-fold linoleic acid (LA) production. After that, they introduced LA isomerase (PAI) into the LA high-yield strain, which afforded a strain producing CLA at 110.6 mg/L. Finally, the authors optimized the medium for CLA production and reached the CLA titer at 132.6 mg/L.
CLA, an important lipid, is marketed as a dietary supplement based on its supposed health benefits, including anti-cancer benefit and bodybuilding aid. Naturally, meat and dairy products contain a small amount of CLA but are insufficient. Hence, metabolic engineering and synthetic biology approaches could be helpful for CLA production. Previous work has shown that an engineered Y. lipolytica could afford 4 g/L CLA production in fermenter [Ref 25]. The CLA production pathway in this work is not new, for they used the same MA12D and PAI in Ref25. The novelty is that they used glycerol as the sole carbon source and overexpressed a key gene, DGA1, for lipid production, resulted in a relatively high CLA titer. As crude glycerol is an important feedstock, this paper would be accepted after essential revisions.
Major points:
1. The author claims the glycerol is the sole carbon source. However, I believe the medium also contains peptone and yeast extract (in Line 287), another two kinds of carbon sources, if the fermentation medium is YPG. Therefore, how much glycerol was converted into CLA? What about the peptone/yeast extract contribution to CLA production?
2. The detection targets were changed in each section, which makes it hard to compare the strain for engineering steps. In Figure 2, the target is lipid content, while in Figure 3 is LA. Even though I understand the aim of the introduction of GPD1 and DGA1 is for lipid production, the LA production/ratio should be added in Figure 2 to link the following engineering step. For Figure 5 of medium optimizations, the criterions changed: LA for nitrogen sources, lipid for C/N ratios, and lipid for CaCl2. If the target is high production of CLA, CLA should be used as the criterion for each optimization in this section.
3. Line 117 to Line 119, introducing of GPD1 increased the lipid titer form 1.86 g/L to 1.95 g/L, I do not think the change is significant. Does the co-expression of GPD1 and DGA1 in the parent strain increase the lipid production?
4. In Figure S3, what are the numbered peaks?
5. The authors owed the increased production of CLA to the strengthening of the copy number and transcription intensity of key recombinant genes (Line244-245). However, neither of the reasons have been validated in this work. If they want to claim these points, they should test CLA production in different copy numbers of the operons and transcription intensities.
Minor points:
1. Please use the full name of LA (Line 20) in the abstract.
2. Please rewrite the sentence from Line 91 to 93 ("During this study, in order to...towards the production of lipids") to make it clear.
Author Response
Major points:
1. The author claims the glycerol is the sole carbon source. However, I believe the medium also contains peptone and yeast extract (in Line 287), another two kinds of carbon sources, if the fermentation medium is YPG. Therefore, how much glycerol was converted into CLA? What about the peptone/yeast extract contribution to CLA production?
Response: We thank the reviewer for alerting us to this problem. It’s our negligence that did not well described the process of strain culture. For lipid and CLA production, the medium is fermentation medium instead of YPG medium. To make it clear, we have revised this part of (page 15 line 309-311).
2. The detection targets were changed in each section, which makes it hard to compare the strain for engineering steps. In Figure 2, the target is lipid content, while in Figure 3 is LA. Even though I understand the aim of the introduction of GPD1 and DGA1 is for lipid production, the LA production/ratio should be added in Figure 2 to link the following engineering step. For Figure 5 of medium optimizations, the criterions changed: LA for nitrogen sources, lipid for C/N ratios, and lipid for CaCl2. If the target is high production of CLA, CLA should be used as the criterion for each optimization in this section.
Response: Thanks to the reviewer’s comments. The linoleic acid content is added in Figure 2. The necessary criterions all changed to “CLA yield” in Figure 5 and we have already modified the corresponding content in 2.4 Optimization of Flask Culture Conditions section.
3. Line 117 to Line 119, introducing of GPD1 increased the lipid titer form 1.86 g/L to 1.95 g/L, I do not think the change is significant. Does the co-expression of GPD1 and DGA1 in the parent strain increase the lipid production?
Response: We thank the reviewer for bringing this problem to our attention. About the comment concerning the co-expression of GPD1 and DGA1 is correct, the result of the experiment confirmed the single-expression of GPD1 can hardly increased the lipid titer where single-expression of DGA1 can obviously promote the accumulation of lipid. We believe the main reason is that glycerol can be fully utilized by the glycerol kinase (Gut1) instead of GPD1, as well as the DGA1 exerts a direct catalytic effect on diacylglycerol (DAG) to form triacylglyceride (TAG). Therefore, overexpression DGA1 represents superior performance and universality in the following study.
4. In Figure S3, what are the numbered peaks?
Response: We thank the reviewer for alerting us to this problem. We have supplemented the names of different ingredients in Figure S3.
5. The authors owed the increased production of CLA to the strengthening of the copy number and transcription intensity of key recombinant genes (Line244-245). However, neither of the reasons have been validated in this work. If they want to claim these points, they should test CLA production in different copy numbers of the operons and transcription intensities.
Response: We thank the reviewer for pointing this out. The sentence is in the discussion part including some deficiencies of this work and future research. The strengthening of the copy number and transcription intensity of key recombinant genes on CLA production is our further research. The used plasmid pINA1312 is a single-copy integrated plasmid, which mean recipient strains carried one integrated copy of pINA1312 plasmid and heterologous genes. The purpose of the study is about the effect of heterologous proteins on CLA production. The expression of the heterologous gene will be driven by the constitutive hp4d promoter, which is almost independent from environmental conditions (pH, carbon and nitrogen sources, peptones), and is able to drive a strong expression in virtually any medium. It consists of regulatory element 4 *UAS1B and core promoter LEU2. Furthermore, there is a linear relationship between the number of UAS1B regulatory elements and the regulation intensity of promoter. The more UAS1B inserted upstream of LEU2, the higher the expression of proteins regulated by promoter[1]. Therefore, changing recombinant plasmid and modification of promoter will be an effective method to strengthen the copy number and transcription intensity of key recombinant genes. To make it clear, some discussion have been modified on page 12 line 245-255.
Minor points:
1. Please use the full name of LA (Line 20) in the abstract.
Response: We thank the reviewer for pointing this out. We have corrected the error on page 1 line 20 and amended the entire manuscript carefully.
2. Please rewrite the sentence from Line 91 to 93 ("During this study, in order to...towards the production of lipids") to make it clear.
Response: We thank the reviewer for alerting us to this problem and have rewritten carefully on page 2 Line 92 to 93.
Submission Date
07 April 2019
Date of this review
15 Apr 2019 02:40:53
References
1. Madzak, C.; Gaillardin, C.; Beckerich, J. M., Heterologous protein expression and secretion in the non-conventional yeast Yarrowia lipolytica: a review. Journal of Biotechnology 2004, 109, (1-2), 63-81.
Reviewer 2 Report
The study aims to grow genetically engineered yeast strains on glycerol, a low-cost substrate, to produce high value trans-10, cis-12 conjugated linoleic acid. To do this, the authors have overproduced diacylglycerol transferase, a desaturase and an isomerase in the oleaginous yeast Yarrowia lipolytica. Improved yields of 110.6 mg/L were obtained.
The project is interesting and the work was carried out well. The manuscript is acceptable if corrections and improvements are made, see below.
Corrections
The structures of the various fatty acids could be shown as a supplementary figure.
Line 85. The LRO1 enzyme should be indicated in Fig. 1
Lines 91 – 93 should read “The copy numbers of pivotal genes, GPD1 or DGA1, were increased, this provides an enhanced driving force towards production of lipids.”
Line 95. In the legend to Fig. 1, “GUT2” in the figure should be explained. This abbreviation also appears without definition in line 232.
Line 96. It is not clear whether the desaturation and double bond migration occur on free fatty acids, or fatty acids esterified as triacyglycerols. There could be a sentence explaining whether this is known or not.
Line 107. Literature is being reviewed in the “Results and Discussion” section. It is difficult to distinguish between previous studies and new results being presented. This could be made clearer by adding phrases like “in this study” or “in this work” e. g. “In this study, to implement the identified improvements, the copy numbers of DGA1 and GPD1 were increased in the Y. lipolytica …………..”
Line 118. WXYL010 has GPD1 and accumulates 1.95 g lipid per litre and WXYL015 has DGA1 and accumulates 2.60 g/L. This is shown in Fig. 2. The text implies that WXYL010 gives 2.6 g/L and WXYL015 gives 1.95 g/L.
Line 131 -134. Again, make a clearer distinction between literature that is being referred to and new results being reported. “In this work, to increase the proportion of unsaturated linoleic acids in lipids, the MAD12D gene was introduced……….”
Line 148. In Fig. 3 “DGAT” should read “DGA1”.
Lines 182 – 183. “…waste oils didn’t been chosen…..” should read “….waste oils were not chosen …..”.
Line 184. Insert the phrase “data not shown”. “Meanwhile, lipid formation/DCW of WXYL037 on glycerol ……..obtained on glucose (data not shown).”
Line 186. “not much enough” should read “not enough”.
Line 219. “flask” not “flashing”.
Line 232. Explain what the GUT2 gene does.
In supplementary Fig. S1, explain what peaks 1, 2, 3, 4, 5 and 6 are in the GL chromatograms.
Author Response
Corrections
The structures of the various fatty acids could be shown as a supplementary figure.
Response: We thank the reviewer for this suggestion and have carefully supplemented Figure S4 in supplementary material.
Line 85. The LRO1 enzyme should be indicated in Fig. 1
Response: Revised as suggested, we have added the LRO1 in Figure 1 according to the reviewer's suggestion.
Lines 91 – 93 should read “The copy numbers of pivotal genes, GPD1 or DGA1, were increased, this provides an enhanced driving force towards production of lipids.”
Response: We thank the reviewer for these comments. Lines 91-93 have been rewritten as suggested.
Line 95. In the legend to Fig. 1, “GUT2” in the figure should be explained. This abbreviation also appears without definition in line 232.
Response: We thank the reviewer for these comments. We have supplemented the explanation of GUT2 in the legend to Figure 1 and page 12 line 240.
Line 96. It is not clear whether the desaturation and double bond migration occur on free fatty acids, or fatty acids esterified as triacyglycerols. There could be a sentence explaining whether this is known or not.
Response: We thank the reviewer for bringing this problem to our attention. The fatty acid isomerase from Propionibacterium acnes (PAI), which catalyzes the isomerization of linoleic acid (LA) to 10,12-CLA[1]. We have added the information on page 2 line88-89.
Line 107. Literature is being reviewed in the “Results and Discussion” section. It is difficult to distinguish between previous studies and new results being presented. This could be made clearer by adding phrases like “in this study” or “in this work” e. g. “In this study, to implement the identified improvements, the copy numbers of DGA1 and GPD1 were increased in the Y. lipolytica…………..”
Response: We thank the reviewer for pointing this comment and added the corresponding phrases on page 3 line 111, page 6 line 139 and page 9 line 164-165.
Line 118. WXYL010 has GPD1 and accumulates 1.95 g lipid per litre and WXYL015 has DGA1 and accumulates 2.60 g/L. This is shown in Fig. 2. The text implies that WXYL010 gives 2.6 g/L and WXYL015 gives 1.95 g/L.
Response: We thank the reviewer for finding this error. We apologize for this mistake, it has now been corrected (Figure 2).
Line 131 -134. Again, make a clearer distinction between literature that is being referred to and new results being reported. “In this work, to increase the proportion of unsaturated linoleic acids in lipids, the MAD12D gene was introduced……….”
Response: Revised as suggested, we amended the entire manuscript carefully and also revised this problem on page 9 line 164-165.
Line 148. In Fig. 3 “DGAT” should read “DGA1”.
Response: Thanks to the reviewer’s comments. We are sorry to have made such a mistake and corrected the mistake in Figure 3.
Lines 182 – 183. “…waste oils didn’t been chosen…..” should read “….waste oils were not chosen …..”.
Response: Thanks to the reviewer’s comments. We have corrected the grammatical mistake on page 11 line 189-190.
Line 184. Insert the phrase “data not shown”. “Meanwhile, lipid formation/DCW of WXYL037 on glycerol ……..obtained on glucose (data not shown).”
Response: Thanks to the reviewer’s comments. We have added the corresponding content on page 11 line 192.
Line 186. “not much enough” should read “not enough”.
Response: Thanks to the reviewer’s comments. We have amended the corresponding content on page 11 line 193.
Line 219. “flask” not “flashing”.
Response: Thanks to the reviewer’s comments. The word has been revised to “flask” on page 11 line 227.
Line 232. Explain what the GUT2 gene does.
Response: Thanks to the reviewer’s comments, we have already added corresponding explanations in the revised manuscript on page 12 line 240.
In supplementary Fig. S1, explain what peaks 1, 2, 3, 4, 5 and 6 are in the GL chromatograms.
Response: Sorry for our mistakes. We have added the explanations of the peaks in Figure S3.
Submission Date
07 April 2019
Date of this review
11 Apr 2019 17:00:52
References
1. Liavonchanka, A.; Hornung, E.; Feussner, I.; Rudolph, M. G., Structure and mechanism of the Propionibacterium acnes polyunsaturated fatty acid isomerase. Proceedings of the National Academy of Sciences of the United States of America 2006, 103, (8), 2576-2581.
Round 2
Reviewer 1 Report
This version looks good. The authors have addressed my concerns, paper is good to go.
Minor point:
Line 356, "Not I" should be "NotI"
Author Response
This version looks good. The authors have addressed my concerns, paper is good to go.
Minor point:
Line 356, "Not I" should be "NotI"
Response: Corrected.